# Dietary Quality Determined by the Healthy Eating Index-2015 and Biomarkers of Chronic Low-Grade Inflammation: A Cross-Sectional Analysis in Middle-to-Older Aged Adults

**DOI:** 10.3390/nu13010222

**Published:** 2021-01-14

**Authors:** Seán R. Millar, Pilar Navarro, Janas M. Harrington, Ivan J. Perry, Catherine M. Phillips

**Affiliations:** 1HRB Centre for Health and Diet Research, School of Public Health, University College Cork, T12 XF62 Cork, Ireland; j.harrington@ucc.ie (J.M.H.); i.perry@ucc.ie (I.J.P.); 2School of Public Health, Physiotherapy and Sports Science, University College Dublin, D04 V1W8 Dublin 4, Ireland; mpilar_ns@hotmail.com

**Keywords:** healthy eating index, inflammation, biomarkers

## Abstract

Low-grade systemic inflammation is associated with a range of chronic diseases. Diet may modulate inflammation and represents a promising therapeutic target to reduce metabolic dysfunction. To date, no study has examined Healthy Eating Index-2015 (HEI-2015) diet score associations with biomarkers of inflammation. Thus, our objective was to assess relationships between the HEI-2015 score and a range of inflammatory biomarkers in a cross-sectional sample of 1989 men and women aged 46–73 years, to test the hypothesis that better dietary quality would be associated with more favourable circulating levels of inflammatory biomarkers. Pro-inflammatory cytokines, adipocytokines, acute-phase response proteins, coagulation factors and white blood cell counts were determined. Correlation and linear regression analyses were used to test HEI-2015 diet score relationships with biomarker concentrations. Higher dietary quality as determined by the HEI-2015 was associated with lower c-reactive protein (CRP) and interleukin 6 concentrations, white blood cell (WBC) counts and its constituents, adjusting for sex and age. Associations with CRP concentrations and WBC counts persisted in the fully adjusted models. No associations with complement component 3, tumour necrosis factor alpha, adiponectin, leptin, resistin or plasminogen activator inhibitor-1 levels were identified. Our data suggest that dietary quality, determined by the HEI-2015 score, in middle-to-older aged adults is associated with inflammatory biomarkers related to cardiometabolic health.

## 1. Introduction

Low-grade systemic inflammation and raised immune activation have been shown to be associated with chronic conditions including type 2 diabetes, cardiovascular disease (CVD), neurodegenerative disease and many cancers [1,2,3,4,5,6]. Consequently, many circulating biomarkers have been evaluated to determine disease risk and their relationships with obesity and certain lifestyle behaviours have been examined [6,7,8]. 

Dietary intake modulates inflammation and represents a promising therapeutic target to reduce metabolic dysfunction and chronic disease [9,10,11,12]. The impact of diet in modulating inflammation is thought to be due to complex interactions between foods and nutrients with bioactive properties [12]. Accordingly, studies have highlighted the importance of characterising the relationship between diet and systemic inflammation through the assessment of dietary patterns, and numerous dietary scores have been developed with the aim of synthesising a large amount of dietary information as a single indicator useful for assessing risk factor–disease relationships [13]. 

The Healthy Eating Index (HEI) is a measure for assessing dietary quality, specifically with regard to the degree which a set of foods align with the Dietary Guidelines for Americans (DGA) [14]. As the DGA are updated every five years, an updated version of the HEI is also released to correspond to each new edition of the DGA; the HEI-2015 is the latest update. Although the HEI-2015 is based on the DGA and, therefore, may not be reliably applied to non-American populations, there is substantial overlap of recommendations, such as consuming more fruit and vegetables and whole grains, while restricting consumption of foods high in sugar and salt, with the food pyramid in Ireland [15]. In addition, versions of the HEI have previously been used to assesses relationships between diet and health outcomes in Ireland and other populations [16,17]. It is important to test the applicability of adapted versions of the HEI in different populations, as the validity of a dietary score depends on the extent to which it is able to distinguish between individuals on relevant health-related intermediate markers [13]. To our knowledge, no study has investigated relationships between diet quality defined by the HEI-2015 score and a wide range of biomarkers of chronic inflammation in a middle-to-older aged population.

Therefore, the aim of the present study was to assess the relationships between the HEI-2015 dietary score and pro-inflammatory cytokines, adipocytokines, acute-phase response proteins, coagulation factors and white blood cells, using a cross-sectional sample of 1989 men and women aged 46–73 years.

## 2. Materials and Methods

### 2.1. Study Population and Setting

The Cork and Kerry Diabetes and Heart Disease Study (Phase II—Mitchelstown Cohort) was a single-centre study conducted between 2010 and 2011. A random sample was recruited from a large primary care centre in Mitchelstown, County Cork, Ireland (Mitchelstown cohort, clinical trials.gov identifier NCT03191227). The Living Health Clinic serves a population of approximately 20,000 white European subjects, with a mix of urban and rural residents. Stratified sampling was employed to recruit equal numbers of men and women from all registered attending patients in the 46–73-year age group. In total, 3807 potential participants were selected from the practice list. Following the exclusion of duplicates, deaths and subjects incapable of consenting or attending appointment, 3051 were invited to participate in the study, and of these, about two-thirds (2047, 49% male) completed the questionnaire and physical examination components of the baseline assessment. Dietary data were available for 1989 subjects. Details regarding the study design, sampling procedures and methods of data collection have been reported previously [18].

Ethics committee approval conforming to the Declaration of Helsinki was obtained from the Clinical Research Ethics Committee of University College Cork (Project identification code: ECM 4 (aa) 02/02/10). A letter signed by the contact GP in the clinic was sent out to all selected participants with a reply slip indicating acceptance or refusal. All participants gave signed informed consent, including permission to use their data for research purposes.

### 2.2. Inflammatory Profiling and Anthropometric Measurements

Study participants attended the clinic in the morning after an overnight fast and blood samples were taken on arrival. Fasting glucose and glycated haemoglobin A_1c_ (HbA_1c_) concentrations were measured in fresh samples by Cork University Hospital Biochemistry Laboratory using standardised procedures. Glucose concentrations were determined using a glucose hexokinase assay (Olympus Life and Material Science Europa Ltd., Lismeehan, Co., Clare, Ireland) and HbA_1c_ levels were measured in the haematology laboratory on an automated high-pressure liquid chromatography instrument Tosoh G7 [Tosoh HLC-723 (G7), Tosoh Europe N.V, Tessenderlo, Belgium]. Serum c-reactive protein (CRP), tumour necrosis factor alpha (TNF-α), interleukin 6 (IL-6), adiponectin, leptin, resistin and plasminogen activator inhibitor-1 (PAI-1) were assessed using a biochip array system (Evidence Investigator; Randox Laboratories, Crumlin, County Antrim, UK). Complement component 3 (C3) was measured by immunoturbidimetric assay (RX Daytona; Randox Laboratories). Total white blood cell (WBC) count, neutrophil, lymphocyte, monocyte, eosinophil and basophil concentrations were determined by flow cytometry technology as part of a full blood count by the Cork University Hospital Haematology Laboratory using fresh blood samples. The neutrophil-to-lymphocyte ratio (NLR) was calculated as neutrophils divided by lymphocytes.

Anthropometric measurements were performed by trained researchers with reference to a standard operating procedures manual. Height was measured with a portable Seca Leicester height/length stadiometer (Seca, Birmingham, UK) and weight was measured using a portable electronic Tanita WB-100MA weighing scale (Tanita Corp., Arlington Heights, IL, USA). The weighing scale was placed on a firm, flat surface and was calibrated weekly. Body mass index (BMI) was calculated as weight in kilograms divided by the square of height in meters. 

### 2.3. Data Collection

A general health and lifestyle questionnaire assessed demographic variables, lifestyle behaviours and morbidity. Information on sex, age, education, prescription anti-inflammatory medication use, smoking status and presence of type 2 diabetes was provided by participants. Physical activity levels were measured using the validated International Physical Activity Questionnaire (IPAQ) [19].

#### 2.3.1. Dietary Assessment

A Food Frequency Questionnaire (FFQ) was used for dietary assessment. Diet was evaluated using a modified version of the self-completed European Prospective Investigation into Cancer and Nutrition (EPIC) FFQ [20], which has been validated extensively in several populations [21]. Adapted to reflect the Irish diet, the 150-item semi-quantitative FFQ used in the current study was originally validated for use in the Irish population using food diaries and a protein biomarker in a volunteer sample [22] and incorporated into the SLÁN Irish National Surveys of Lifestyle Attitudes and Nutrition 1998, 2002 and 2007 [23,24,25]. The FFQ also was validated using a seven-day weighed food record completed in another Irish study (Lifeways Cross-generational Study), with reasonable agreement for fat, carbohydrate and their components, and with lower agreement for protein [26].

The average medium serving of each food item consumed by participants over the last 12 months was converted into quantities using standard portion sizes. Food item quantity was expressed as (g/day) and beverages as (mL/day). The daily intake of energy and nutrients was computed from FFQ data using a tailored computer programme (FFQ Software Version 1.0; developed by the National Nutrition Surveillance Centre, School of Public Health, Physiotherapy and Sports Science, University College Dublin, Belfield, Dublin 4, Ireland), which linked frequency selections with the food equivalents in McCance and Widdowson Food Tables [27].

#### 2.3.2. HEI-2015 Score

The HEI-2015 is a measure of overall diet quality that measures alignment with the 2015–2020 DGA [16,28]. The HEI-2015 contains 13 components which are scored on a density basis out of 1000 calories, with the exception of fatty acids, which is a ratio of unsaturated to saturated fatty acids [14]. Total fruits, whole fruits, total vegetables, greens and beans, total protein-containing foods and seafood and plant proteins scored 5 in the highest consumption and 0 in the lowest consumption. The highest consumption of three components, including whole grains, dairy and fatty acids (ratio of poly- and monounsaturated fatty acids to saturated fatty acids), is scored as 10 and the lowest consumption is scored as 0. Four components (refined grains, sodium, added sugars and saturated fats) scored 10 in the lowest consumption and 0 in the highest consumption [14]. Component scores are summed to yield a total score ranging from 0 to 100, with a higher score indicating greater adherence to the DGA. In our sample, scores ranged from 21–62.

### 2.4. Classification and Scoring of Variables

Categories of education included ‘some primary (not complete)’, ‘primary or equivalent’, ‘intermediate/group certificate or equivalent’, ‘leaving certificate or equivalent’, ‘diploma/certificate’, ‘primary university degree’ and ‘postgraduate/higher degree’. These were collapsed and recoded into a dichotomous variable: ‘primary education only’ (finished full-time education at age 13 years or younger) and ‘intermediate or higher’. Type 2 diabetes was determined as a fasting glucose level ≥ 7.0 mmol/L or a HbA_1c_ level ≥ 6.5% (≥48 mmol/mol) [29] or by self-reported diagnosis. 

Smoking status was defined as follows: (i) never smoked, i.e., having never smoked at least 100 cigarettes (five packs) in their entire life; (ii) former smoker, i.e., having smoked 100 cigarettes in their entire life and do not smoke at present; and (iii) current smoker, i.e., smoking at present. These definitions were the same as those used in the SLÁN National Health and Lifestyle Survey [30]. A binary variable was then created: ‘never/former smoker’ or ‘current smoker’. Physical activity was categorised as low, moderate and high levels of activity using the IPAQ. This was then recoded as a dichotomous variable: ‘moderate/high’ or ‘low’ physical activity. 

### 2.5. Statistical Analysis 

Descriptive characteristics were examined according to sex and HEI-2015 score quartiles. Categorical features are presented as percentages and continuous variables are shown as a mean (plus or minus one standard deviation) or a median and interquartile range for skewed data. Differences were analysed using a Pearson’s chi-square test, Student’s *t*-test or a Mann Whitney U test. Trend relationships were examined using a Jonckheere test, a linear-by-linear chi-square or an ANOVA. Correlations between individual dietary score components, the HEI-2015 score and biomarker concentrations were examined using Spearman’s rank-order correlation.

The HEI-2015 score was standardised and skewed biomarker data were log-transformed for linear regression analysis to examine associations between the HEI-2015 score and biomarker levels. Three models were run. The first model was used to test crude associations. A second model adjusted for sex and age. The final model adjusted for sex, age, education, smoking, physical activity, total energy intake, anti-inflammatory medication use, type 2 diabetes and BMI.

Data analysis was conducted using Stata SE Version 13 (Stata Corporation, College Station, TX, USA) for Windows. For all analyses, a *p* value (two-tailed) of less than 0.05 was considered to indicate statistical significance.

## 3. Results

### 3.1. Descriptive Characteristics

Characteristics of the study population for the full sample and according to sex are presented in Table 1. Significant differences between the sexes were noted for education, use of anti-inflammatory medications, type 2 diabetes, physical activity, BMI and the HEI-2015 dietary score, with male participants having poorer diet quality than females. Sex differences were also observed for all biomarker levels, with the exception of lymphocyte and basophil concentrations. Table 2 shows characteristics of the study population according to HEI-2015 dietary score quartiles. Lower HEI-2015 scores indicate poorer dietary quality, whereas higher scores indicate a healthier diet. Subjects with poorer diet quality (quartile 1 compared to quartile 4) were more likely to be male, have lower educational levels, to be current smokers, to report lower physical activity and total energy intake levels and had higher (lower for adiponectin) concentrations of inflammatory and thrombotic biomarkers than did those who consumed higher quality diets.

### 3.2. Correlation Analysis and Linear Regression

In correlation analyses (Table 3), significant relationships between individual dietary score components and biomarkers were observed for C3, CRP, IL-6, TNF-α, adiponectin, leptin, WBC, neutrophils, lymphocytes, the NLR, monocytes and eosinophils, with total fruits, whole fruits, green and beans, fatty acids, added sugars and saturated fats showing the greatest number of significant relationships. No associations for dairy, total protein foods, refined grains or sodium were noted with any inflammatory biomarker. Weak but significant inverse correlations between the HEI-2015 dietary score and biomarkers were seen for concentrations of CRP, IL-6, TNF-α, WBC, neutrophils, lymphocytes, the NLR, monocytes, eosinophils and basophils. Adiponectin levels were positively correlated with the HEI-2015 score.

Table 4 shows linear regression models demonstrating relationships between standardised HEI-2015 scores and inflammatory and thrombotic biomarkers. In the analyses, which were adjusted for sex and age, significant associations between the HEI-2015 score and biomarkers were observed for CRP, IL-6, WBC, neutrophil, lymphocyte, monocyte and eosinophil concentrations. In the fully adjusted models, significant inverse relationships remained for CRP, WBC and neutrophil levels.

## 4. Discussion

In this study of 1989 men and women, we examined the HEI-2015 dietary score associations with markers of chronic low-grade inflammation and raised immune activation. Dietary quality, as determined by the HEI-2015 score, was inversely associated with CRP and IL-6 concentrations and WBC counts. The associations with CRP levels and WBC counts remained significant following adjustments for a range of potential confounders. These findings suggest reduced systemic inflammation as a potential biological mechanism linking a higher quality, healthy diet with beneficial health effects. 

Chronic low-grade inflammation is a major contributor to chronic conditions including metabolic syndrome, type 2 diabetes and CVD [31]. Low-grade systemic inflammation may also promote cancer development by increasing the levels of reactive oxygen and nitrogen (“free radicals”), which can damage DNA. In addition, inflammatory cytokines are thought to activate transcription factors that promote cancer progression through changes in signalling pathways that promote cell proliferation and resistance to cell death [32]. The impact of diet in modulating inflammation is thought to be due to complex interactions between foods and nutrients with bioactive properties [12]. Our examination of individual dietary score components revealed associations with inflammatory markers (C3, CRP, IL-6, TNF-α, adiponectin, leptin, WBC, neutrophils, lymphocytes, the NLR, monocytes and eosinophils), with total and whole fruits, vegetables, greens and beans, fatty acids, added sugars and saturated fats showing the greatest number of significant relationships. It is worth noting that no associations for dairy, total protein foods, refined grains or sodium were observed with any inflammatory biomarker. In agreement with our findings, several studies have shown that vegetable- and fruit-based diets are inversely associated with inflammatory markers, whereas meat-based diets, poor in vegetable and omega-3 fatty acid intake and high in refined carbohydrates, added sugar, saturated and trans fatty acids tend to be positively associated with biomarkers of inflammation [33,34,35]. To our knowledge, no study thus far has examined relationships between the HEI-2015 score and circulating inflammatory biomarkers. Although we are the first to report on this, future research replicating our findings in other populations is warranted.

Previous studies have highlighted the importance of characterising the relationship between diet and systemic inflammation through an assessment of dietary patterns. Fung et al. found beneficial effects of adhering to the Dietary Approaches to Stop Hypertension (DASH) in reducing inflammation in a 24-year follow-up of women from the Nurses’ Health Study [36]. These findings were supported in a meta-analysis of randomised controlled trials, which suggest that the beneficial effects of the DASH diet on reducing chronic disease risk are due not only to reductions in blood pressure, but also due to improvements in inflammatory biomarker levels [37]. More recently, the Dietary Inflammatory Index (DII^®^) was developed specifically to measure the inflammatory potential of diet based on the overall inflammatory properties of dietary components, with a meta-analysis showing that individuals with the highest DII scores, and thus, the most pro-inflammatory diet had a 36% increased risk of CVD incidence and mortality relative to those with the lowest DII score [38].

Although both the DASH and DII scores have been validated against inflammatory biomarkers in previous research [12,31,39], fewer studies have examined the HEI score as a marker of systemic inflammation and the findings have been inconsistent. Data from the National Health and Nutrition Examination Survey (1998–2004) in a sample of 13,811 men and women suggested an inverse relationship between HEI scores and serum CRP [40]. However, in an analysis of 690 women from the Nurses’ Health Study, Fung et al. found no association between the HEI and concentrations of CRP or IL-6 [41]. Recent research, using a sample of 133 community-dwelling older adults, also found HEI-2010 composite dietary scores not to be significantly associated with decreased inflammation characterised by CRP, TNF-α and IL-6 levels or greater anti-oxidant capacity [42]. However, it is worth pointing out the smaller sample sizes, sex differences, age ranges and different HEI scores examined in the latter studies may account for the disparity and lack of association. 

The HEI dietary score is a well-validated metric of dietary quality outlined by the DGA, with greater adherence to a higher quality diet demonstrating significant inverse associations with various chronic diseases. A meta-analysis of 15 studies (*n* = 1,020,632) found the HEI-2005 and HEI-2010 scores to be significantly associated with a reduction in all-cause mortality, CVD, cancer and type 2 diabetes (*p* < 0.00001) [43]. A prospective analysis of 12,413 participants aged 45–64 years from the Atherosclerosis Risk in Communities Study found that compared with participants in the lowest HEI-2015 score quintile, participants in the highest quintile had a 16% lower risk of incident CVD, 32% lower risk of CVD mortality and an 18% lower risk of all-cause mortality [44]. While our findings suggest reduced systemic inflammation as a potential mechanism which might underly such associations, examination of other chronic non-communicable disease biomarkers may provide further mechanistic insight. Indeed, a recent investigation of HEI and HEI-2010 scores and chronic disease risk among low-income adolescents found that the HEI, but not the HEI-2010 score, was associated with biomarkers of chronic disease risk including total and low-density lipoprotein cholesterol, percent body and abdominal fat and impaired glucose tolerance [45]. The authors suggest that differences in the underlying structure of the scores, in particular dietary quality versus quantity, are responsible for the disparity in predictive ability of chronic disease risk. While most of the HEI-2010 features were retained in the HEI-2015 score, they differ in some respects, reflecting changes between the 2010 and 2015 DGA. Most noteworthy is the “empty calories component”, which has been replaced by “Added sugars” and “Saturated fats”; moreover, alcohol intake has been removed. Additionally, from a life course cardiovascular epidemiology perspective, it is important to bear in mind that age-related changes in these cardiovascular risk factors occur. Thus, future studies would benefit from examining HEI-2015-biomarker associations across the life course.

This study has several strengths. With the elderly population growing [46], it is to be expected that the number of patients with non-communicable diseases will increase. Modifications in certain lifestyle behaviours and adopting a healthier diet may help prevent against chronic conditions, and this may be of particular importance to older adults. This research is the first to compare HEI-2015 relationships with a range of markers of chronic low-grade inflammation and raised immune activation in a middle-to-older aged population, and thus, our study has examined the largest number of biomarkers using an iteration of the HEI. Research on dietary indices is important for public health, as studies on these can provide better insights into disease causation. Other strengths include the large number of middle-to-older aged study participants, equal representation by sex (49% male) and the use of validated questionnaires to collect data.

Despite these strengths, several limitations should be noted. The cross-sectional study design, which precludes drawing conclusions regarding the temporal direction of relationships, limits inference with respect to causality. This should be considered in light of decades of work on the association between diet and serum lipids, which suggest that relationships may be discernible by only using longitudinal data [47]. In addition, the use of self-reported questionnaires is subject to potential inaccuracies. Thus, it should be noted that as a structured dietary assessment technique, the FFQ is less precise than 24-hour recall and food records; furthermore, as a method based on long-term memory, it can introduce recall and reporting biases [48]. However, this approach has been shown to provide valid estimates of food intake in older adults [49]. Furthermore, although the HEI-2015 is a well-validated and evolving tool for the evaluation of dietary quality [50], it is based on the DGA and, therefore, may not be as reliably applicable to non-American populations. Nevertheless, there is a substantial overlap of recommendations (e.g., consuming more fruit and vegetables and whole grains, while restricting consumption of foods high in sugar and salt) with the food pyramid in Ireland [15]. In addition, a recent large multi-ethnic cohort study (*n* > 215,000), which investigated the predictive validity of the HEI-2015 score, supported the idea that a high-quality diet positively influences biologic pathways involved in chronic disease etiology across different ethnic groups [51]. However, in our middle-to-older aged sample, HEI-2015 scores ranged from 21–62. As homogeneity of diet will increase the likelihood of not detecting a true relationship between diet and inflammatory markers, we acknowledge that where the study population and their eating habits/food culture and preferences are more diverse, there may be a greater ability to detect relationships between diet and markers of chronic low-grade inflammation, especially if foods consumed are the ones with more anti/pro-inflammatory effects and/or are consumed in large amounts. Finally, and related to the previous points, the generalisability of our findings may be limited. Our data were collected from a single primary care-based sample, which may not be representative of the general population. However, Ireland represents a generally ethnically homogeneous population [52]. In addition, previous research suggests that approximately 98% of Irish adults are registered with a GP and that, even in the absence of a universal patient registration system, it is possible to perform population-based epidemiological studies that are representative using our methods [53].

## 5. Conclusions

In conclusion, the results from this research suggest an inverse association between the HEI-2015 dietary score and circulating CRP concentrations and WBC counts in middle-to-older aged adults. A more favourable inflammatory status may be a potential mechanism linking higher quality diet and reported health benefits of a healthy diet. An examination of dietary scores, such as the HEI-2015, provides a more holistic way of looking at habitual diets in comparison to examining individual macronutrients or selected food items. This approach may also be more translatable to public health messaging. Furthermore, our data suggest that consumption of certain dietary components of the HEI-2015 score (fruit, vegetables, legumes, added sugars and fats) are of particular importance with respect to inflammatory status. A better understanding of the relationships between diet and biomarkers of health is needed, with a view to informing public health nutrition policy and promotion of healthy eating to improve dietary quality and ultimately overall health and well-being.

## Figures and Tables

**Table 1 nutrients-13-00222-t001:** Characteristics of the study population—full sample and stratified by sex.

Variable	Full Sample (*n* = 1989)	Males (*n* = 973)	Females (*n* = 1016)	*p* Value
Age (median)	59.0 (54.5–64.0)	59.0 (55.0–64.0)	59.0 (54.0–63.0)	0.888
Primary education only (%)	521 (27.8)	298 (32.2)	223 (23.6)	<0.001
On anti-inflammatory medications (%)	331 (17.0)	204 (21.5)	127 (12.7)	<0.001
Type 2 diabetes (%)	177 (8.9)	113 (11.6)	64 (6.3)	<0.001
Current smoker (%)	283 (14.4)	139 (14.4)	144 (14.3)	0.939
Low-level physical activity (%)	906 (48.2)	381 (42.2)	525 (53.6)	<0.001
BMI [kg/m^2^] (mean)	28.6 ± 4.7	29.1 ± 4.1	28.0 ± 5.1	<0.001
Energy intake, kcal (mean)	2036.0 ± 812.6	2058.8 ± 808.0	2014.3 ± 816.8	0.222
HEI-2015 score (mean)	39.5 ± 7.0	38.7 ± 7.0	40.3 ± 7.0	<0.001
C3 [mg/dL] (mean)	135.84 ± 24.7	134.16 ± 22.4	137.4 ± 26.7	0.003
CRP [ng/mL] (median)	1.35 (0.98–2.30)	1.32 (0.96–2.13)	1.38 (0.99–2.46)	0.039
IL-6 [pg/mL] (median)	1.78 (1.19–2.91)	1.92 (1.27–3.08)	1.68 (1.12–2.71)	<0.001
TNF-α [pg/mL] (median)	5.87 (4.89–7.29)	6.09 (5.06–7.48)	5.90 (4.76–7.15)	<0.001
Adiponectin [ng/mL] (median)	4.74 (2.92–7.54)	3.19 (2.21–4.92)	6.65 (4.44–9.61)	<0.001
Leptin [ng/mL] (median)	1.95 (1.09–3.16)	1.58 (0.84–2.60)	2.24 (1.27–4.27)	<0.001
Resistin [ng/mL] (median)	5.05 (3.92–6.89)	4.88 (3.82–6.50)	5.22 (3.99–6.96)	0.002
PAI-1 [ng/mL] (mean)	27.37 ± 12.5	29.04 ± 13.1	25.70 ± 11.8	<0.001
WBC [10^9^/L] (median)	5.70 (4.80–6.80)	5.90 (5.10–7.10)	5.50 (4.60–6.50)	<0.001
Neutrophils [10⁹/L] (median)	3.12 (2.52–3.93)	3.28 (2.66–4.15)	2.98 (2.38–3.76)	<0.001
Lymphocytes [10⁹/L] (median)	1.74 (1.42–2.14)	1.73 (1.41–2.14)	1.76 (1.44–2.15)	0.51
NLR (median)	1.78 (1.40–2.29)	1.85 (1.48–2.39)	1.67 (1.32–2.19)	<0.001
Monocytes [10⁹/L] (median)	0.50 (0.40–0.62)	0.54 (0.44–0.68)	0.45 (0.37–0.56)	<0.001
Eosinophils [10⁹/L] (median)	0.17 (0.11–0.26)	0.19 (0.12–0.28)	0.16 (0.10–0.24)	<0.001
Basophils [10⁹/L] (median)	0.03 (0.02–0.04)	0.03 (0.02–0.04)	0.03 (0.02–0.04)	0.702

Abbreviations: C3: complement component 3; CRP: c-reactive protein; HEI: Healthy Eating Index; IL-6: interleukin 6; TNF-α: tumour necrosis factor alpha; PAI-1: plasminogen activator inhibitor 1; WBC: white blood cell counts; NLR: neutrophil-to-lymphocyte ratio. Numbers and percentages may vary as some variables have missing values. *p* values determined from a Mann–Whitney U, Pearson’s chi-square or *t*-test.

**Table 2 nutrients-13-00222-t002:** Characteristics of the study population according to the HEI-2015 dietary score quartiles.

Variable	HEI-2015 Score Quartiles (*n* = 1989)				
	Q1	Q2	Q3	Q4	*p* Trend
Age (median)	59.0 (55.0–64.0)	59.0 (55.0–64.0)	59.0 (55.0–64.0)	54.0 (55.0–63.0)	0.252
Male (%)	300 (57.7)	251 (51.1)	208 (42.8)	214 (43.5)	<0.001
Primary education only (%)	153 (31.0)	137 (29.7)	113 (24.6)	118 (25.8)	0.025
On anti-inflammatory medications (%)	79 (15.5)	80 (26.7)	84 (17.5)	88 (18.3)	0.22
Type 2 diabetes (%)	44 (8.5)	45 (9.2)	40 (8.2)	48 (9.8)	0.602
Current smoker (%)	93 (18.0)	84 (17.4)	57 (11.9)	49 (10.1)	<0.001
Low-level physical activity (%)	270 (55.4)	222 (47.7)	217 (47.1)	197 (42.1)	<0.001
BMI [kg/m^2^] (mean)	28.8 ± 4.8	28.6 ± 4.6	28.6 ± 4.8	28.3 ± 4.4	0.126
Energy intake, kcal (mean)	1876.7 ± 645.4	1928.0 ± 709.0	1979.5 ± 739.1	2368.0 ± 1019.8	<0.001
C3, mg/dL (mean)	136.72 ± 25.2	135.17 ± 23.8	136.81 ± 25.8	134.59 ± 23.9	0.345
CRP, ng/mL (median)	1.43 (0.99–2.40)	1.31 (0.98–2.13)	1.36 (0.96–2.35)	1.28 (0.92–2.18)	0.019
IL-6, pg/mL (median)	1.88 (1.26–3.12)	1.68 (1.16–2.71)	1.83 (1.20–3.03)	1.74 (1.15–2.76)	0.048
TNF-α, pg/mL (median)	6.22 (5.16–7.37)	5.93 (4.83–7.34)	5.77 (4.64–7.08)	6.01 (4.89–7.34)	0.039
Adiponectin, ng/mL (median)	4.38 (2.87–6.75)	4.79 (2.94–7.54)	4.83 (3.05–7.75)	4.96 (2.96–7.75)	0.012
Leptin, ng/mL (median)	1.91 (1.10–3.02)	1.89 (1.05–3.01)	2.07 (1.14–3.37)	1.88 (1.05–3.00)	0.593
Resistin, ng/mL (median)	5.21 (4.00–6.90)	4.97 (3.85–6.55)	4.98 (3.95–6.64)	5.01 (3.88–6.67)	0.312
PAI-1, ng/mL (mean)	28.24 ± 12.7	27.42 ± 11.90	26.03 ± 11.21	27.73 ± 14.06	0.248
WBC, 10^9^/L (median)	5.90 (5.10–7.20)	5.70 (4.70–6.50)	5.60 (4.70–6.50)	5.50 (4.60–6.50)	<0.001
Neutrophils, 10⁹/L (median)	3.37 (2.70–4.17)	3.07 (2.51–3.92)	3.09 (2.49–3.87)	2.96 (2.36–3.76)	<0.001
Lymphocytes, 10⁹/L (median)	1.76 (1.45–2.21)	1.75 (1.40–2.15)	1.73 (1.41–2.09)	1.73 (1.41–2.12)	0.055
NLR (median)	1.86 (1.49–2.36)	1.76 (1.37–2.27)	1.77 (1.40–2.25)	1.72 (1.31–2.25)	0.004
Monocytes, 10⁹/L (median)	0.52 (0.43–0.66)	0.50 (0.40–0.62)	0.49 (0.39–0.61)	0.47 (0.38–0.59)	<0.001
Eosinophils, 10⁹/L (median)	0.18 (0.12–0.26)	0.19 (0.12–0.27)	0.16 (0.10–0.25)	0.16 (0.11–0.25)	0.007
Basophils, 10⁹/L (median)	0.03 (0.02–0.04)	0.03 (0.02–0.04)	0.03 (0.02–0.04)	0.03 (0.02–0.04)	0.05

Abbreviations: C3: complement component 3; CRP: c-reactive protein; IL-6: interleukin 6; TNF-α: tumour necrosis factor alpha; PAI-1: plasminogen activator inhibitor 1; WBC: white blood cell counts; NLR: neutrophil-to-lymphocyte ratio. *p* values for trend determined using a Jonckheere test, a linear-by-linear chi-square test or an ANOVA.

**Table 3 nutrients-13-00222-t003:** Spearman correlation coefficients between individual dietary score components, the HEI-2015 dietary score and inflammatory and thrombotic biomarkers.

**Biomarker**	**Total Fruits**		**Whole Fruits**		**Total Vegetables**		**Greens and Beans**		**Whole Grains**		**Dairy**		**Total Protein Foods**	
	**rho**	***p***	**rho**	***p***	**rho**	***p***	**rho**	***p***	**rho**	***p***	**rho**	***p***	**rho**	***p***
C3, mg/dL	−0.012	0.593	−0.043	0.059	−0.012	0.602	−0.025	0.266	−0.061	**0.008**	−0.039	0.088	0.005	0.827
CRP, ng/mL	−0.032	0.155	−0.048	**0.035**	−0.025	0.272	−0.023	0.314	−0.072	**0.002**	0.034	0.135	0.003	0.883
IL-6, pg/mL	−0.083	**<0.001**	−0.102	**<0.001**	−0.029	0.209	−0.058	**0.01**	−0.045	**0.048**	0.028	0.213	0.027	0.237
TNF-α, pg/mL	−0.046	**0.043**	−0.061	**0.007**	−0.050	**0.028**	−0.063	**0.006**	−0.003	0.878	0.016	0.485	0.026	0.259
Adiponectin, ng/mL	0.132	**<0.001**	0.161	**<0.001**	0.058	**0.011**	0.114	**<0.001**	−0.042	0.063	−0.001	0.977	−0.015	0.497
Leptin, ng/mL	0.049	**0.031**	0.060	**0.009**	0.029	0.205	0.061	**0.007**	−0.052	**0.021**	0.007	0.756	0.010	0.65
Resistin, ng/mL	0.000	0.983	0.004	0.85	−0.034	0.134	0.004	0.862	−0.016	0.478	0.010	0.66	−0.019	0.409
PAI-1, ng/mL	−0.018	0.43	−0.031	0.167	−0.038	0.093	−0.021	0.366	−0.015	0.508	−0.035	0.119	0.000	0.985
WBC, 10^9^/L	−0.126	**<0.001**	−0.150	**<0.001**	−0.093	**<0.001**	−0.078	**0.001**	−0.033	0.146	−0.003	0.905	−0.004	0.873
Neutrophils, 10⁹/L	−0.122	**<0.001**	−0.146	**<0.001**	−0.093	**<0.001**	−0.086	**<0.001**	−0.016	0.492	0.000	0.991	−0.011	0.644
Lymphocytes, 10⁹/L	−0.034	0.133	−0.040	0.079	−0.019	0.409	0.006	0.782	−0.053	**0.02**	−0.002	0.921	0.019	0.413
NLR	−0.078	**0.001**	−0.091	**<0.001**	−0.061	**0.007**	−0.065	**0.004**	0.029	0.195	−0.003	0.906	−0.020	0.388
Monocytes, 10⁹/L	−0.139	**<0.001**	−0.152	**<0.001**	−0.087	**<0.001**	−0.089	**<0.001**	−0.027	0.228	−0.020	0.378	−0.009	0.688
Eosinophils, 10⁹/L	−0.064	**0.005**	−0.085	**<0.001**	−0.053	**0.021**	−0.054	**0.017**	−0.041	0.071	−0.015	0.499	−0.013	0.578
Basophils, 10⁹/L	−0.037	0.1	−0.036	0.115	−0.035	0.119	−0.004	0.85	−0.044	0.05	−0.020	0.369	−0.018	0.418
**Biomarker**	**Seafood and Plant Proteins**		**Fatty Acids**		**Refined Grains**		**Sodium**		**Added Sugars**		**Saturated Fats**		**HEI-2015 Score**	
	**rho**	***p***	**rho**	***p***	**rho**	***p***	**rho**	***p***	**rho**	***p***	**rho**	***p***	**rho**	***p***
C3, mg/dL	0.018	0.416	−0.008	0.709	0.001	0.969	0.033	0.149	0.043	0.057	−0.024	0.292	−0.025	0.281
CRP, ng/mL	0.003	0.893	−0.052	**0.023**	0.041	0.069	0.032	0.165	0.028	0.216	−0.070	**0.002**	−0.065	**0.004**
IL−6, pg/mL	−0.007	0.769	−0.070	**0.002**	0.015	0.508	0.029	0.208	0.069	**0.002**	−0.051	**0.025**	−0.057	**0.012**
TNF-α, pg/mL	0.005	0.83	−0.043	0.059	−0.034	0.134	0.004	0.87	0.049	**0.032**	−0.056	**0.015**	−0.048	**0.033**
Adiponectin, ng/mL	0.001	0.967	0.056	**0.014**	−0.016	0.474	0.022	0.336	−0.117	**<0.001**	0.049	**0.03**	0.057	**0.011**
Leptin, ng/mL	0.046	**0.042**	0.009	0.693	0.006	0.776	0.011	0.62	−0.042	0.065	−0.008	0.711	0.013	0.574
Resistin, ng/mL	0.003	0.888	−0.004	0.857	−0.036	0.111	0.027	0.235	−0.041	0.071	−0.020	0.386	−0.023	0.312
PAI−1, ng/mL	0.034	0.136	−0.033	0.148	0.004	0.849	−0.021	0.352	0.027	0.24	−0.014	0.533	−0.032	0.162
WBC, 10^9^/L	−0.028	0.216	−0.074	**0.001**	0.009	0.679	0.026	0.25	0.053	**0.019**	−0.073	**0.001**	−0.125	**<0.001**
Neutrophils, 10⁹/L	−0.044	0.054	−0.083	**<0.001**	0.009	0.701	0.026	0.247	0.052	**0.023**	−0.083	**<0.001**	−0.128	**<0.001**
Lymphocytes, 10⁹/L	0.024	0.296	−0.020	0.38	0.001	0.981	0.009	0.698	0.016	0.474	−0.035	0.121	−0.049	**0.032**
NLR	−0.042	0.064	−0.051	**0.025**	0.004	0.856	−0.002	0.945	0.023	0.312	−0.041	0.074	−0.072	**0.002**
Monocytes, 10⁹/L	−0.035	0.123	−0.086	**<0.001**	0.017	0.466	0.005	0.835	0.066	**0.003**	−0.045	**0.048**	−0.118	**<0.001**
Eosinophils, 10⁹/L	−0.013	0.579	−0.049	**0.032**	−0.011	0.642	0.018	0.417	0.055	**0.016**	−0.049	**0.03**	−0.075	**0.001**
Basophils, 10⁹/L	0.003	0.887	−0.017	0.45	−0.013	0.576	0.010	0.653	0.002	0.916	−0.027	0.239	−0.050	**0.028**

Abbreviations: C3: complement component 3; CRP: c-reactive protein; IL-6: interleukin 6; TNF-α: tumour necrosis factor alpha; PAI-1: plasminogen activator inhibitor 1; WBC: white blood cell counts; NLR: neutrophil-to-lymphocyte ratio. Values are presented as Spearman correlation coefficients (rho) between individual dietary score components, the HEI-2015 score and inflammatory and thrombotic biomarkers among the Mitchelstown Cohort (*n* = 1989). Significant *p*
**highlighted**.

**Table 4 nutrients-13-00222-t004:** Linear regression analysis of the associations between the HEI-2015 dietary score and inflammatory and thrombotic biomarkers (*n* = 1989).

	Model 1			Model 2			Model 3		
	β	S.E.	*p*	β	S.E.	*p*	β	S.E.	*p*
C3	−0.390	0.565	0.49	−0.563	0.567	0.321	−0.407	0.593	0.492
Log CRP *	−0.040	0.016	**0.013**	−0.043	0.016	**0.008**	−0.035	0.017	**0.035**
Log IL-6 *	−0.049	0.017	**0.004**	−0.035	0.017	**0.033**	−0.024	0.018	0.183
Log TNF-α *	−0.015	0.008	0.062	−0.010	0.008	0.196	−0.016	0.009	0.073
Log Adiponectin *	0.040	0.016	**0.012**	0.002	0.014	0.904	0.002	0.015	0.876
Log Leptin *	0.006	0.021	0.759	−0.020	0.020	0.317	−0.013	0.019	0.486
Log Resistin *	−0.011	0.010	0.262	−0.014	0.010	0.153	−0.013	0.011	0.217
PAI-1	−0.299	0.285	0.294	−0.108	0.285	0.706	0.225	0.315	0.474
Log WBC *	−0.035	0.006	**<0.001**	−0.030	0.006	**<0.001**	−0.013	0.006	**0.045**
Log Neutrophils *	−0.042	0.008	**<0.001**	−0.035	0.008	**<0.001**	−0.016	0.008	**0.047**
Log Lymphocytes *	−0.020	0.007	**0.007**	−0.021	0.007	**0.004**	−0.007	0.008	0.356
Log NLR *	−0.022	0.009	**0.016**	−0.014	0.009	0.113	−0.008	0.010	0.409
Log Monocytes *	−0.043	0.007	**<0.001**	−0.031	0.007	**<0.001**	−0.011	0.008	0.149
Log Eosinophils *	−0.049	0.014	**0.001**	−0.039	0.014	**0.006**	−0.021	0.016	0.177
Log Basophils *	−0.024	0.013	0.058	−0.025	0.013	0.051	−0.007	0.014	0.624

Abbreviations: C3: complement component 3; CRP: c-reactive protein; IL-6: interleukin 6; TNF-α: tumour necrosis factor alpha; PAI-1: plasminogen activator inhibitor 1; WBC: white blood cell counts; NLR: neutrophil to lymphocyte ratio. Model 1: univariate. Model 2: adjusted for sex and age. Model 3: adjusted for sex, age, education, smoking, physical activity, total energy intake, anti-inflammatory medication use, type 2 diabetes and BMI. Unstandardised β coefficients and standard errors (S.E.) are shown. Significant *p*
**highlighted**. * Log-transformed.

## Data Availability

The data used and analysed for the purpose of this study are available from the corresponding author on reasonable request.

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
