# Peer review of "Dietary Quality Determined by the Healthy Eating Index-2015 and Biomarkers of Chronic Low-Grade Inflammation: A Cross-Sectional Analysis in Middle-to-Older Aged Adults"

_nutrients, 2021, doi:10.3390/nu13010222_

Round 1

Reviewer 1 Report

The current manuscript describes a cross-sectional analysis of dietary and inflammatory data. It is effectively described, although I feel that elements of the Discussion could be further expanded.

  1. Tables - please consider removal of all horizontal gridlines in tables, except for those above and below the column heading and below the final row. The data are already aligned, so the multiple gridlines tend to make the content less appealing to read.
  2. Discussion - possibly within the strengths and limitations section, I feel that the authors need to discuss their dietary data collection approach and analysis further. In particular, the strengrhs and limitations of data collection using FFQs should be discussed. Further from this, the relevance of HEI estimates of dietary ideal are based on US dietary guidelines. Considerations of how these differ/align from the guidelines in the Republic of Ireland should be presented, alongside consideration of whether the data collection approach adequately aligns with HEi-2015 scoring approaches.

Author Response

Point by point response

Reviewer 1

The current manuscript describes a cross-sectional analysis of dietary and inflammatory data. It is effectively described, although I feel that elements of the Discussion could be further expanded.

Point 1: Tables - please consider removal of all horizontal gridlines in tables, except for those above and below the column heading and below the final row. The data are already aligned, so the multiple gridlines tend to make the content less appealing to read.

Response 1: The tables have now been formatted according to the reviewer's suggestion.

Point 2. Discussion - possibly within the strengths and limitations section, I feel that the authors need to discuss their dietary data collection approach and analysis further. In particular, the strengths and limitations of data collection using FFQs should be discussed. Further from this, the relevance of HEI estimates of dietary ideal are based on US dietary guidelines. Considerations of how these differ/align from the guidelines in the Republic of Ireland should be presented, alongside consideration of whether the data collection approach adequately aligns with HEI-2015 scoring approaches.

Response 2: Thank you for raising these important points. We have added information to the Limitations section of the manuscript stating that as a structured dietary assessment technique, the FFQ is less precise than 24-hour recall and food records; furthermore, as a method based on long-term memory it can introduce recall and reporting biases, but that this approach has been shown to provide valid estimates of food intake in older adults. We also note that although the HEI-2015 is a well-validated and evolving tool for the evaluation of dietary quality, it is based on the “Dietary Guidelines for Americans”, and so may not be reliably applied to non-American populations. Nevertheless, there is substantial overlap of recommendations (e.g., consuming more fruit and vegetables, whole grains, while restricting consumption of foods high in sugar and salt) with the food pyramid in Ireland. In addition, recent research which investigated the predictive validity of HEI-2015 score supports the idea that a high-quality diet positively influences biological pathways involved in chronic disease etiology across different ethnic groups (Panizza, C.E.; Shvetsov, Y.B.; Harmon, B.E.; Wilkens, L.R.; Le Marchand, L.; Haiman, C.; Reedy, J.; Boushey, C.J. Testing the predictive validity of the healthy eating index 2015 in the multiethnic cohort: is the score associated with a reduced risk of all cause and cause-specific mortality? Nutrients 2018, 10, 45).

Reviewer 2 Report

This study explored the association between HEI-2015 score and various inflammatory biomarkers in a cross-sectional sample of Irish adults. As low-grade systemic inflammation is regarded as an important determinant in a range of chronic disease, the topic of this study is interesting and the sample size is fairy large, although it recruited only one center that this study would be meaningful addition in this field. However, several concerns about methodology has been risen that comments warrant consideration.

Health Eating Index 2015 (HEI-2015)

As it is well introduced in the Introduction and Method, HEI-2015 is a measure of overall diet quality. However, it was developed from 2015-2020 Dietary Guideline for Americans. Why did authors choose this tool for Irish adults? HEI score was originally created to monitor the nutritional status for Americans based on the dietary data for US population. Dietary patterns analysis or dietary scores can be a better indicator than a single nutrient indicator in order to explore relationship between diet and health outcome. But it is used with caution when it applies to other countries with different dietary characteristics. Please explain the rationale in the Introduction

Distribution of HEI-2015 dietary score

What is the distribution of HEI-2015 dietary score in this population? Is it similar with USA adults? Before further analysis, authors should fully explain the main dietary variable.

Also understanding the components of HEI-2015 can help analyze and interpret the data. For example, authors figure out which components were more correlated or associated with low grade inflammatory variables.

Spearman’s rank-order correlation and Linear regression analysis

According to Statistical analysis, HEI-2015 scores were log-transformed due to skewness. I guess that because of skewness, authors used spearman’s rank-order correlation (non-linear relationship), instead of Pearson correlation. But why did you use linear regression analysis?

Usually, dietary score is not normally distributed because it is sum of score in each category, I doubt linear regression analysis for each inflammation indicator is appropriate although it is log-transformed.

Discussion

Author mainly discussed: 1) chronic low-grade inflammation and its association with chronic disease, 2) HEI score and its association with chronic disease. In particular, authors discussed the different versions of HEI and claimed that this study was the first study to use HEI-2015 and explore its association with low-grade inflammation.

However, HEI-2015 is just one of indicators to evaluate overall dietary quality. Thus, I suggest adding more discussion of the impact or association of nutrient levels/ food levels on the low-grade inflammation and then discuss how this study add more explanation beyond that regarding the relationships between diet and low-grade inflammation.

Author Response

Point by point response

Reviewer 2

Comments and Suggestions for Authors

This study explored the association between HEI-2015 score and various inflammatory biomarkers in a cross-sectional sample of Irish adults. As low-grade systemic inflammation is regarded as an important determinant in a range of chronic disease, the topic of this study is interesting and the sample size is fairly large, although it recruited only one center that this study would be meaningful addition in this field.

Authors: We are glad the reviewer finds the topic of our investigation to be of interest and that our study represents a meaningful contribution to the area.

However, several concerns about methodology has been risen that comments warrant consideration.

Point 1: Health Eating Index 2015 (HEI-2015). As it is well introduced in the Introduction and Method, HEI-2015 is a measure of overall diet quality. However, it was developed from 2015-2020 Dietary Guideline for Americans. Why did authors choose this tool for Irish adults? HEI score was originally created to monitor the nutritional status for Americans based on the dietary data for US population. Dietary patterns analysis or dietary scores can be a better indicator than a single nutrient indicator in order to explore relationship between diet and health outcome. But it is used with caution when it applies to other countries with different dietary characteristics. Please explain the rationale in the Introduction.

Response 1: Thank you for this comment. We have added information to the Introduction section stating that although the HEI-2015 is based on the “Dietary Guidelines for Americans” and, therefore, may not be reliably applied to non-American populations, there is substantial overlap of recommendations, such as consuming more fruit and vegetables and whole grains, while restricting consumption of foods high in sugar and salt, with the food pyramid in Ireland. In addition, versions of the HEI have previously been used to assesses relationships between diet and health outcomes in Ireland and other populations. We also state that it is important to test the applicability of adapted versions of the HEI in different populations, as the validity of a dietary score depends on the extent to which it is able to distinguish between individuals on relevant health-related intermediate markers. These points are also discussed in the Strengths and Limitations section of the manuscript as requested by Reviewer 1.

Point 2: Distribution of HEI-2015 dietary score

What is the distribution of HEI-2015 dietary score in this population? Is it similar with USA adults? Before further analysis, authors should fully explain the main dietary variable.

Response 2: The distribution of the HEI-2015 score is shown in Table 1, both for the full sample and stratified by sex. The scores ranged from 21–62 and this information has now been included in the manuscript (Section: 2.3.2. HEI-2015 score). It is important to note that our sample was made up of a middle- to older-aged population of subjects aged 46–73 years, so a direct comparison of score distribution and range with an American population is perhaps not helpful. However, we have discussed in the Limitations section of the manuscript that homogeneity of diet will increase the likelihood of not detecting a true relationship between diet and inflammatory markers, and that we acknowledge that where the study population and their eating habits/food culture and preferences are more diverse, then there may be a greater ability to detect relationships between diet and markers of chronic low-grade inflammation, especially if foods consumed are the ones with more anti/pro-inflammatory effects and/or are consumed in large amounts.

Point 3: Also understanding the components of HEI-2015 can help analyze and interpret the data. For example, authors figure out which components were more correlated or associated with low grade inflammatory variables.

Response 3: We agree and have included individual dietary score component correlations with biomarkers in Table 3. We report that “In correlation analyses (Table 3), significant relationships between individual dietary score components and biomarkers were observed for C3, CRP, IL-6, TNF-α, adiponectin, leptin, WBC, neutrophils, lymphocytes, the NLR, monocytes and eosinophils, with total fruits, whole fruits, vegetables, green and beans, fatty acids, added sugars and saturated fats showing the greatest number of significant relationships. No associations for dairy, total protein foods, refined grains or sodium were noted with any inflammatory biomarker”.

Point 4: Spearman’s rank-order correlation and Linear regression analysis

According to Statistical analysis, HEI-2015 scores were log-transformed due to skewness. I guess that because of skewness, authors used spearman’s rank-order correlation (non-linear relationship), instead of Pearson correlation. But why did you use linear regression analysis? Usually, dietary score is not normally distributed because it is sum of score in each category, I doubt linear regression analysis for each inflammation indicator is appropriate although it is log-transformed.

Response 4: Thank you for asking us to clarify this. The HEI-2015 score was not log-transformed but was standardised. This is a useful technique when a predictor variable has a very large scale, as this leads to regression coefficients of a very small order of magnitude. Linear regression was used to test relationships between the HEI-2015 and biomarkers to account for potential confounders. Skewed biomarker data (which are the outcome variables) were log-transformed, and this is the methodology that has been used in previous research examining dietary score relationships with biomarkers (Piccand, E.; Vollenweider, P.; Guessous, I.; Marques-Vidal, P. Association between dietary intake and inflammatory markers: Results from the CoLaus study. Public health nutrition 2019, 22, 498-505; Monfort-Pires, M.; Folchetti, L.D.; Previdelli, A.N.; Siqueira-Catania, A.; de Barros, C.R.; Ferreira, S.R.G. Healthy Eating Index is associated with certain markers of inflammation and insulin resistance but not with lipid profile in individuals at cardiometabolic risk. Applied Physiology, Nutrition, and Metabolism 2014, 39, 497-502; Sanjeevi, N.; Lipsky, L.M.; Nansel, T.R. Cardiovascular biomarkers in association with dietary intake in a longitudinal study of youth with type 1 diabetes. Nutrients 2018, 10, 1552; Millar, S.R.; Harrington, J.M.; Perry, I.J.; Phillips, C.M. Protective lifestyle behaviours and lipoprotein particle subclass profiles in a middle-to older-aged population. Atherosclerosis 2020, 314, 18-26).

Point 5: Discussion

Author mainly discussed: 1) chronic low-grade inflammation and its association with chronic disease, 2) HEI score and its association with chronic disease. In particular, authors discussed the different versions of HEI and claimed that this study was the first study to use HEI-2015 and explore its association with low-grade inflammation. However, HEI-2015 is just one of indicators to evaluate overall dietary quality. Thus, I suggest adding more discussion of the impact or association of nutrient levels/ food levels on the low-grade inflammation and then discuss how this study add more explanation beyond that regarding the relationships between diet and low-grade inflammation.

Response 5: As previously mentioned, we have examined individual HEI-2015 dietary score component relationships with biomarkers in Table 3 and these additional findings are now discussed in the Results and Discussion section (paragraph 2) of the manuscript.

Round 2

Reviewer 2 Report

Authors addressed most comments in a good manner. 

There are minor things to be addressed. Regarding the linear regression analysis, all biomarkers (inflammatory variables) are log-transformed? If so, please indicate it at the footnote. 

Another thing is that did you compare the correlation coefficients of each component of Spearman with Pearson analysis? How was it? If HEI-2015 score is standardized, I assmum that Pearson correlation coefficient can be a better predictor. 

Author Response

Point by point response

Reviewer 2

Authors addressed most comments in a good manner.

Thank you.

Point 1: There are minor things to be addressed. Regarding the linear regression analysis, all biomarkers (inflammatory variables) are log-transformed? If so, please indicate it at the footnote.

Response 1: Log-transformed biomarkers are shown with the prefix “Log” in Table 4; e.g., “Log CRP”. However, we have also included this information in the footnote of Table 4 (Page 11, line 253).

Point 2: Another thing is that did you compare the correlation coefficients of each component of Spearman with Pearson analysis? How was it? If HEI-2015 score is standardized, I assume that Pearson correlation coefficient can be a better predictor.

Response 2: Thank you for asking us to clarify this. We used correlation analysis to assess significant univariate relationships and the strength of correlations between individual dietary score components, the HEI-2015 score and biomarkers. We used Spearman’s Rank-Order Correlation as it is a more robust test than a Pearson Product-Moment Correlation and is more appropriate to use when one or both variables do not follow a normal distribution. It should be noted that neither individual dietary score components or HEI-2015 scores were standardised in correlation analyses. Standardising a variable does not make it follow a normal distribution (unlike log-transforming a variable). Also, the Spearman’s correlation between a standardised score and biomarker will be the same as the Spearman’s correlation between a non-standardised score and biomarker. We only standardised the HEI-2015 score in linear regression analyses as this is a useful technique when a predictor variable has a very large scale, which may lead to regression coefficients of a very small order of magnitude. For further clarification, we have changed wording in the Statistical analysis section of the manuscript to read “The HEI-2015 score was standardised and skewed biomarker data were log- transformed for linear regression analysis to examine associations between the HEI-2015 score and biomarker levels” (Page 5, Line 188).
